# Peer review of "Sema4C Is Required for Vascular and Primary Motor Neuronal Patterning in Zebrafish"

_cells, 2022, doi:10.3390/cells11162527_

Round 1

Reviewer 1 Report

The article by Sheng J. Et al., entitled “Sema4C is required for vascular and primary motor neuronal patterning in zebrafish”, focuses on the role of Semaphorin4C (Sema4C) in zebrafish embryos’ vascular and neural development. The authors observe that Sema4C is expressed in both nervous system and intersegmental vessels (ISVs) using whole embryo in situ hybridization. Moreover, authors report that Sema4C knockdown performed with Morpholino and CRISPR/Cas9 leads to morphological abnormalities in both ISVs and primary motor neurons of zebrafish embryos. Finally, observing the rescue of the phenotype after Sema4C mRNA reintroduction, authors demonstrate that the described abnormalities in neuronal and vascular components were specifically due to Sema4C. Given these evidences, authors suggest that Sema4C is a dual guiding factor in vascular and neural development. Overall, I find this study complete, original and the data well presented. Therefore, I believe that this article is eligible for publication.

Minor revisions:

·         Some microscopy images miss the magnification bar. In particular Figure2 panels A’ and B’; Figure 4A, Figure 5C (specifically move the bar from the second image to the first) and figure 6C.

·         I suggest to explain in the figure legend the mimic diagrams from the microscopy images of both images 3 and 4.

·         The English of the results section (section 3) should be carefully checked since there are some mistakes in verbal tenses.

·         The first sentence of paragraph 3.3 (lines 1 and 2) should be reformulated.

·         The third sentence of paragraph 2.6 (lines 4 and 5) should be reformulated and the rescue phenotype should be better explained.

Author Response

Some microscopy images miss the magnification bar. In particular Figure2 panels A’ and B’; Figure 4A, Figure 5C (specifically move the bar from the second image to the first) and figure 6C.

Response: We thank the reviewer for pointing it out and we added it in the revised version.

  • I suggest to explain in the figure legend the mimic diagrams from the microscopy images of both images 3 and 4.

Response: We thank the reviewer for the professional advices and we detailed it in the new version.

  • The English of the results section (section 3) should be carefully checked since there are some mistakes in verbal tenses.

Response: We thank the reviewer for the professional advices and we revised it in the new version.

  • The first sentence of paragraph 3.3 (lines 1 and 2) should be reformulated.

Response: We thank the reviewer for pointing it out and we revised it in the new version.

  • The third sentence of paragraph 2.6 (lines 4 and 5) should be reformulated and the rescue phenotype should be better explained.

Response: We thank the reviewer for the professional advices and we revised it in the revised version.

Reviewer 2 Report

The MS entitled “Sema4C is required for vascular and primary motor neuronal patterning in zebrafish “by Sheng et alii, investigates the role of the sema4C gene in controlling vascular and neuronal development in zebrafish embryos. This MS shows a detailed assessment by microscopy of sema4C expression during zebrafish development, which leads to investigating the loss of function of this gene in vascular and neuronal differentiation using morpholinos and CRISPR. Results presented in this MS are novel and robust (showing the right controls), and the statistical analysis supports their claims. I therefore recommend this MS for publication in Cells if the minor concerns are assessed appropriately.

Minor concern:

At the beginning of their the MS, authors claim (verbatim) “The study might serve as a ground work for future researches to determine the molecular mechanisms of axon guidance factors in vascular development, the processes involved in neurovascular communication, and understanding the disease associated with vascular abnormalities and nerves development defects.” Which refers to the usefulness of their results to treat/investigate human diseases. This is, to my understanding, an exaggerated assertion that has to be softened before being considered for publication in Cells. At the end of their MS, authors suggest that their results may be helpful for studying in depth the mechanisms involved in nerve and vascular development, however, which is a more appropriate conclusion for this sort of study.

Author Response

At the beginning of their the MS, authors claim (verbatim) “The study might serve as a ground work for future researches to determine the molecular mechanisms of axon guidance factors in vascular development, the processes involved in neurovascular communication, and understanding the disease associated with vascular abnormalities and nerves development defects.” Which refers to the usefulness of their results to treat/investigate human diseases. This is, to my understanding, an exaggerated assertion that has to be softened before being considered for publication in Cells. At the end of their MS, authors suggest that their results may be helpful for studying in depth the mechanisms involved in nerve and vascular development, however, which is a more appropriate conclusion for this sort of study.

Response: We thank the reviewer for the professional advices. We revised it in the new version.